# Salience, Credibility and Legitimacy in a Rapidly Shifting World of Knowledge and Action

**David W. Cash [1,\*] and Patricio G. Belloy [1,2]** 

[1] John W. McCormack Graduate School of Policy and Global Studies, University of Massachusetts Boston, Boston, MA 02125, USA; patricio.belloy001@umb.edu

[2] Institute of Economics, Austral University of Chile, Valdivia, Los Rios 5090000, Chile

\* Correspondence: david.cash@umb.edu; Tel.: +1-617-287-5511

**Abstract:** We are in a rapidly changing world where new dynamics are stressing the knowledge-action landscape: a greater understanding that cross-scale interactions are critical; increasing pressure to more fully address issues of equity in sustainable development challenges; rapidly transforming digital technologies; and the emergence of a "post-truth world". These stressors are ripening at a time in which there is increased urgency in linking knowledge to action to solve some of the earth's most pressing human-environment problems. This paper explores to what degree one model of knowledge-action may be useful in the face of these stressors. This model relies on co-production of knowledge across boundaries, and the importance of knowledge in meeting criteria of salience, credibility and legitimacy. Tentative explorations suggest utility of this model in responding to the changing knowledge-action landscape.

**Keywords:** knowledge; action; boundary work; cross-scale; equity; digital transformation; post-truth; climate

## 1. Introduction

Over 18 months in 2015 to 2016, citizens filled dozens of community rooms, churches and schools in the US city of Boston and met with government officials, university analysts and consultants to analyze the intersection of race, income and the local impacts of global climate change [1]. Half a world away in rural Tanzania, teams of scientists from national and international bodies collaborated with local communities to co-produce knowledge about the changing climate that integrated scientific and indigenous knowledge in forms that communities could use to adapt to global climate change at the local level [2]. Parallel to these efforts, the Intergovernmental Panel on Climate Change (IPCC) was preparing its "Special Report: Global Warming of 1.5 °C." [3]. The effort engaged decision-makers and scientists from all over the world, from developed and developing countries, and across dozens of disciplines.

These three efforts—in different cultures, engaging different institutions and stakeholders, utilizing different resources and situated at different scales—represent a decades-long sea change in how the interface of science and policy, knowledge and action, has evolved [4–9].

Each of these three cases manifest a more nuanced understanding that the relationship between science and decision-making is "situated within cultural and institutional contexts that shape how individuals and organizations engage with one another, and co-construct rules, norms, and practices that shape interactions between science and society" [9] (p. 336). Each of these cases demonstrate the growing notion that co-production of knowledge can result in greater utilization of knowledge to drive action [10,11]. Each of these cases is consistent with recent research that "suggests that potential users [of scientific or technical information] are more likely to trust new knowledge, and may therefore be

more willing to act on it, when from their perspective, it meets three criteria: saliency, credibility, and legitimacy" [10] (p. 109, emphasis in original). Each of these cases show the importance of "boundary work" across multiple boundaries—from science to action, across disciplines, across public and private spheres, across issue areas, and across scales [12–17].

In the last several decades, researchers have studied themes of salience, credibility, and legitimacy, sustainability science, co-production, boundary work, and other forms of linking knowledge and action in more intentional and strategic ways and have used them prescriptively to guide the implementation of a range of efforts around the world. Naturally, these efforts are not without critique, and limitations of the framework have been identified, including counteracting effects, interactions between salience, credibility and legitimacy and trade-offs to meet the criteria [18–20]; changing perceptions about salience, credibility and legitimacy over time [21]; or sudden lost in achieved validity because of later interaction with non-trusted partners [22]. In the last decade, moreover, several emergent or growing dynamics in complex socio-ecological systems provide stressors to these developing systems. Four such new or growing dynamics include: (1) increasing pressure to link knowledge and action from the global through local levels; (2) prioritizing issues of equity and distributive justice more centrally in solutions about environmental and sustainability questions; (3) rapid expansion of digital technologies, artificial intelligence (AI) and social media, and resultant impacts on ways of producing and consuming knowledge; and (4) degradation of trust in formal and traditional science—and expertise in general—most prominently driven in the U.S. context by President Donald Trump and other conservative leaders. The following sections explore the impact of these dynamics and provide tentative insights into how the framework of salience, credibility and legitimacy and co-production may be able to mitigate or even leverage these stressors/dynamics in the effort to better link knowledge and action.

## 2. Knowledge and Action across Scales

Each of the above cases shows a deepening grasp by scientists and decision-makers that understanding cross-scale interactions are critical to both comprehending complex socio-ecological systems and to crafting solutions to such problems [23,24]. This is not a new insight and, indeed, some of the earliest theories of managing the commons acknowledged the impact of the interaction of higher jurisdictions (such as national governments or international bodies like the United Nations) on the provision of collective public goods through the actions of individuals and households [25,26]. In recent years, attempts to address the challenges and opportunities of cross-scale interactions have weaved them into the fabric of the research and practice of knowledge and action [10,27–35].

Given this evolution, it is not a surprise that the United Nations Sustainable Development Goals (SDGs) resolution of 2015 identified the importance of the "local" in solving the global challenges of poverty, health, education, environment and peace [36]. In fact, the final resolution contains the term "local" ten times and includes phrases like, "We will work with local authorities and communities . . . " (p. 9).

Likewise, the 2015 Paris Agreement on Climate Change was the first global agreement to explicitly identify "sub-national" jurisdictions as critical players in the effort to decrease greenhouse gas emissions and meet Nationally Determined Contributions (NDCs) [37]. Following the agreement, the commitments and actions made by sub-national actors (such as cities) has increasingly been a critical piece of the climate solution implementation puzzle. Organizations such as C40 and the Rockefeller Foundation's 100 Resilient Cities have been at the forefront on laws, regulations and programs at the city level, and of the piloting and modeling of best practices and capacity-building. C40 is a network of megacities—representing over 700 million citizens and one quarter of the global economy—that are committed to taking bold climate action. The network supports cities to reach the goals of the Paris Agreement through effective collaboration by sharing knowledge and best practices [38]. The 100 Resilient Cities (100RC) initiative launched in 2013 by the Rockefeller Foundation as part of the 100-year anniversary of its founding, devoting $100 million to its implementation. The initiative supports

cities to become more resilient and prepare for the most pressing challenges of the 21st century [39]. Both efforts have explicitly connected their actions to not only national policy but to international governance as well.

This linking across scales is not only embraced for planning, policy and implementation, but also for connecting science from the global to the local levels. At the completion of its deliberations in 2016, the UN Secretary General's Scientific Advisory Board report recommended, "Indeed, what is really needed is an efficient science-policy-society interface that will create and make use of a holistic framework including a diversity of stakeholders, from government, civil society, indigenous peoples and local communities, businesses, academia, and research organizations" [40] (p.17). Embedded in this framework, needs to be an integration of local knowledge and global knowledge: the knowledge of communities situated in place and knowledge created at the national and global levels: "In this sustainable development era, science should incorporate all valuable inputs, including from indigenous and local knowledge systems" (p. 23).

In the framework of salience, credibility and legitimacy, a multi-level world creates obvious challenges—what knowledge is salient, credible or legitimate to a farmer, city councilor or tribal leader, may be far different than what knowledge is salient, credible or legitimate to a national leader, UN scientist, or multi-national company's corporate board. In a world where demand for linking across scales is growing (e.g., 900 mayors attended the Paris climate negotiations), assuring that the criteria of salience, credibility and legitimacy are met across scales, and thus creating trust [10] is that much more important.

## 3. Knowledge, Action and Equity

Each of the cases highlighted at the beginning of this paper (Boston, Tanzania and the IPCC) also illustrate the growing effort to harness knowledge to more directly address issues of justice and equity in the context of sustainability, and to develop institutions specifically designed to accomplish this. This has become increasingly urgent in light of four considerations over the last several decades.

First, the ethical dimensions of equity have been central to discussions about sustainability, but that theme was put into sharp relief in 2015 with the publication of Pope Francis' encyclical, *Laudato Si', In Care of Our Common Home* [41]. As spiritual leader to the world's 1.2 billion Catholics, the Pope has a unique authority to assert the links between poverty, environmental degradation and moral responsibility: "Today, however, we have to realize that a true ecological approach *always* becomes a social approach; it must integrate questions of justice in debates on the environment, so as to hear *both the cry of the earth and the cry of the poor*" (p. 35, emphasis in original). The Vatican's nod to the principles of human ecology is also a sign of change that acknowledges another profound reality: "the relationship between human life and the moral law, which is inscribed in our nature and is necessary for the creation of a more dignified environment" (p. 115). This encyclical, of course, is only one of many spiritual and moral expressions of the increasing focus on equity at the nexus of knowledge and action for sustainability.

Second, populations with existing vulnerabilities (whether categorized by economic status, colonial legacy, race, health status, etc.) are most negatively affected by climate-related impacts [42–44], facing greater risks of health and property damage and less capacity to adapt and recover [45,46]. Members of these communities are the most likely to have to re-locate and become climate refugees [47,48]. This vulnerability has been further highlighted during the COVID-19 pandemic as these communities have again suffered disproportionate transmission, infection and death [49,50].

Third, recovery efforts themselves may contribute to greater inequities post-disaster [51,52]. More privileged populations are better able to take advantage of government-sponsored recovery efforts and have greater access to private sources of insurance [53].

Fourth, historically disadvantaged populations are often unable to take advantage of inherent opportunities in sustainability transformations [54,55]. In the United States, for example, this manifests in several ways. Deployment of clean energy, cost-saving technology such as energy efficiency retrofits,

residential solar energy and electric vehicles has not proportionately expanded into low-income communities compared to middle- and upper-income communities [56–58]. Likewise, the rapid expansion of clean energy jobs in the United States has also not made proportionate inroads in low-income communities that have higher levels of unemployment [59,60].

Issues of equity are characterized by the allocation of resources (what is salient to whom), fairness in distribution, justice, and who wins and who loses. These highly politically charged ideas are often concerning for scientists, who fear that engaging in the discussion may be threatening to their credibility. As Matson et al. [10] note, "[m]any researchers resist the notion that science and politics are intermingled, but when knowledge is salient to the interests of actors in society—whether as an invention that might be bought and sold or a discovery that advances some agendas while undermining others—then knowledge becomes power." (p. 122) The buzz-phrase "knowledge is power" and its implication for equity is one of the driving assumptions of the emerging fields/movements of energy democracy [61], environmental justice [62] and climate justice [63]. While these are topics about which equitable allocation of resources are critical, they also speak to issues of transparency. Who has a seat at the table? How fair are the processes that create the knowledge that drives decision-making? How do the processes of knowledge (co)production empower particular voices and knowledges? In short, these are questions about the legitimacy of how knowledge is created and used.

## 4. Knowledge and Action in a Digital World

The digital revolution has transformed markets, consumerism, knowledge production, knowledge dissemination and the news media. It has spawned new sectors, launched some of the largest multi-national corporations in the world and driven new fields of research. Scientific collaboration is arguably easier and the sharing of information and data more seamless. More citizens, globally, can have access to information at their fingertips and often can leapfrog traditional technologies to advance a range of societal goals, including sustainability goals. For example, numerous examples of distributed solar energy deployment, electric vehicle use, residential energy efficiency and many other programs have been enabled by the smart grid, smart technology and ever-growing penetration of and access to digital devices [64–66].

Given that digital platforms and associated technologies are broadly distributed, they have the potential to make knowledge creation more transparent, and even the collection and analysis of data can extend easily beyond the realm of academia or government scientists. In some domains they have certainly expanded citizen science—from eBird identification to web-based fish counting at fish ladders—and opened possibilities of monitoring and reporting of air and water pollution and other environmental quality tracking that would not have been possible or would have been much more resource intensive in the past. Vast amounts of satellite and other earth monitoring data are now readily available through government and NGO website-based datasets.

The accessibility of digital information may imply that the boundaries between science and citizens and between science and policy makers are more porous and more easily crossed, and that digitally-driven co-production facilitates enhanced salience, credibility and legitimacy. But is this the case?

Because of both the relatively recent ascendency of the digital revolution and the rapid evolution of new technologies, relatively little research that explores the impact on knowledge-action or science-policy dimensions exists. Lember et al. [67] review current findings and "bring together the core conceptual elements . . . on co-production (direct interaction, motivation, resources, and shared decision-making) and digital technologies (sensing, communication, processing, and actuation) and show how the latter can affect co-production/-creation processes." (p. 1671) Not surprisingly, the outcomes are varied and mixed. See Table 1.

**Table 1.** Potential positive and negative impacts of digital technologies on co-production/co-creation.

| | Sensing | Communication | Processing | Actuation |
|---|---|---|---|---|
| **Interaction** | Increases interaction where deliberate inputs by citizens are necessary. Diminishes the perceived need for interaction with citizens. | Allows swifter and broader exchange of information. Digital interaction diminishes physical interaction. | Allows a more effective selection of specific target groups to interact with or manipulate through nudging. | Increases human-to-machine interaction. Reduces human-to-human interaction or cuts out human interaction altogether. |
| **Motivation** | Allows a level of personalization of services that increases motivation. Diminishes motivation through fear and surveillance and information overload. | Increases motivation by lower threshold, better evidence and more entertainment. Decreases motivation by crowding out intrinsic motives and threatening privacy. | With the aid of communication technology, increases personalization and thus motivation. With the aid of communication technology, enables more effective nudging that decreases the will to co-produce. | Increases motivation as new opportunities for co-production emerge. Decreases motivation as automation leads to disengagement with the service process. |
| **Resources** | Generates data that can be used to increase the quality and scope of co-production/co-creation. | Allows the mobilization of resources from citizens on a far wider scale. Enables hidden privatization. | Generates new resources, which can be used to increase or decrease interaction, motivation and shared decision-making. | Lowers the time and effort needed to co-produce. Increases need for technical skills and strengthens existing inequalities. |
| **Decision-making** | Data from sensing allows citizens to become part of the decision-making process. Data from sensing allow data owners to exclude citizens from decision-making. | Empowers citizens through a more open process and improved knowledge. Diminishes the need for shared decision-making, by allowing governments to manipulate and citizens to self-organize more effectively. | With the aid of communication technology, supports a more shared decision-making process. With the aid of actuation technology, supports both more open and more closed types of decision-making. | Control can be decentralized through adaptive decision-making. Control can be centralized, making programmed decisions without any direct input from citizens. |

Source: Lember et al. (2019), p. 1676.

In general, various dimensions of co-production can be enhanced by digital technologies through more interactions, new ways to collaborate, more access to more stakeholders, greater efficiencies, lower costs, increased options for communication and sharing of information, and improved access to analytical capabilities. Alternately, digital technologies can have adverse impacts on co-production, reducing the need for human-to-human contact, privileging technological know-how, requiring resources for hardware and software and therefore exacerbating existing inequalities, increasing automation over human engagement, enhancing centralization of decision-making, increasing the possibility of manipulation of data or information (see the next section), increasing the ability for censorship, or by generating answers rooted in datasets/big data modeled by decisions that hide unfair historic practices against historically disempowered groups [68]. Lember et al. [67] note that, "[d]igital technologies have the potential to strengthen the participatory element in this latest paradigm. However, in parallel, we may be witnessing the emergence of a 'fourth paradigm', one in which decision-making in public services interacts with citizens only indirectly, through nudging, or bypasses them altogether, basing decisions on complex algorithms and collective data." (p. 1681) As the rapid change in the digital knowledge production and dissemination evolves, how can institutions that encourage salience, credibility and legitimacy influence which of these two paths is followed?

## 5. Knowledge and Action in a "Post-Truth" World

As noted above, part of our framework focuses on credibility—the traditional "coin of the realm" for scientific research, in which peer review, accepted methodology and agreed-upon-standards are used to assure that science is believable. The framework further explores and operationalizes the tradeoffs and complementarities between and among credibility, salience and legitimacy, accepting the notion that science is created in a political and social context in which information may or may not be

salient to decision-makers and processes may or may not confer legitimacy to scientific activities. This model also accepts that there is not an "objective" credibility—that is, standards of credibility may differ depending on the audience—and credibility can be contested.

Many argue, however, that a fundamental and qualitative shift has occurred. In the last decade or more (at least in the United States and arguably in countries like Hungary, Turkey, or Brazil), some claim we have entered an era of "post-truth", in which the credibility of science itself, not just particular scientific endeavors, is questioned [69,70]. This erosion is often seen in specific areas such as those outlined by the work of Naomi Oreskes and Erik Conway, who chart the decades-long history of tobacco and fossil fuel companies undermining credibility of science in those arenas [71]. Clark and Harley [72] concur, "[a]n even deeper cause for concern highlighted by the coproduction perspective is that when researchers persist and do create knowledge that threatens powerful interests vested in the status quo, they often induce push back, personal attacks, or outright disinformation campaigns. Ongoing efforts to undermine research-based knowledge on the role of fossil fuels in driving the climate crisis and the role of junk food in driving the malnutrition crisis are well-known examples." (p. 46).

These specific cases have broadened to a more general distrust of science and is manifest not only in issues that are interest-based from a political economy perspective (like tobacco and fossil fuels), but also in more general issues in the public sphere (for example, about vaccinations and the coronavirus). Given its recent history, little formal analysis is currently available, but it seems clear that various communications, management, regulatory and legal efforts by President Donald Trump and his administration's executive agencies have resulted in both issue-specific and broader questioning of science and expertise. Driven by direct communications in popular media (social and traditional), notions of "fake news" and "hoaxes" often frame the discussion of policy-relevant scientific debates [73,74]. Reducing funding within agencies such and the National Oceanic and Atmospheric Administration and the Environmental Protection Agency, attacking the science of international bodies like the IPCC, disbanding the Global Health Security and Biodefense unit in the National Security Council, and diminishing scientific advisory boards through executive agencies signal a decrease in reliance on science for decision-making [75–78]. Regulatory changes have diminished the input of science in permitting and other regulatory processes [79–81].

The salience, credibility and legitimacy framework assumes that these characteristics matter. Science itself being challenged raises fundamental questions about how we understand knowledge and action. In a "post-truth" world, what role does credibility play and what are the implications for credibility, salience and legitimacy in understanding the knowledge-action nexus, and in prescriptive strategy?

## 6. Hypotheses about Salience, Credibility, and Legitimacy in a World of New Stressors

Over the last two decades, scholars and practitioners at the boundary of knowledge and action have found the framework useful that posits that users of information "are more likely to trust new knowledge . . . when . . . it meets three criteria: saliency, credibility, and legitimacy" [10]. The recent shifting landscape of knowledge-action is characterized by these four stressors that force the question of whether this framework can also be useful in the face of these fluctuating dynamics.

### 6.1. Cross-Scale Dynamics

Of the four stressors, cross-scale dynamics is the least "new". In fact, some of the early work on salience, credibility and legitimacy focused on cross-scale dynamics as one of the fundamental challenges facing human sustainability issues, and different levels of governance and knowledge production contain boundaries that often need to be crossed [14]. However, the greater focus on the importance of linking local to global knowledge and action, and the increase in "bottom up" action in the face of national or global inaction, brings this stressor more to the fore.

Examples of global agreements (e.g., SDGs and the Paris Agreement) that explicitly identify the importance of the local level and municipalities that link their actions to global agreements (e.g., C40 and the 100 Resilient Cities) are innovative example of bridging boundaries. In practice, these efforts attempt to institutionalize cross-scale interactions in ways that will enhance salience, credibility and legitimacy. For C40 cities (which represent approximately 10% of the global population), the salience of addressing climate change at the local level is key to the effort to "share knowledge". In each of these examples, efforts are being made to create processes of co-producing knowledge across scales and relying on academic and intergovernmental processes that can assure credibility. In short, as framed by Matson et al., [10] these efforts are designed to increase the trust in scientific knowledge used by local actors, and therefore increase the probability of "meaningful, measurable and sustainable action".

There may be temporal, spatial, and institutional mismatches across scales between local and global problems. Local actors are sometimes confined within short-run concerns of their jurisdictions and within other institutional settings. Like in the case of industrial relocation, municipalities may have a legitimate say in local decision that may produce regional or global impacts [82]. On the other hand, global entities or powerful regional actors, due to resourcefulness, mandate or other reasons, may also be over-driving the agenda, affecting trust across scales.

These dynamics suggest a testable hypothesis: greater explicit calls to institutionalize linking local to global scales will result in increased salience, credibility and legitimacy of knowledge co-produced across scales and therefore greater connection between knowledge and action from the global to the local level. One way to test this hypothesis is to look at how countries are adjusting institutional arrangements to meet their NDCs and respond to the requirements established by the Paris Convention in terms of collaboration across scales. Among different cases to examine, the Government of Colombia is working with local authorities to integrate the climate agenda and SDGs into local-level planning and budgetary processes in all departments of the country [83]. Likewise, Kenya's National Climate Change Council is including representation from marginalized populations, civil society and academia as part of their strategy to decentralize climate action and incorporate subnational authorities in NDC planning efforts [84]. Sweden's institutional framework, for example, depends heavily on county administrative boards and their collaboration with regional and local stakeholders to design and implement regional climate and energy strategies. Costa Rica, on the other side, conducted an external audit of their NDC elaboration process to ensure participation and inclusiveness, convening government departments at all levels, national civil society organizations and international NGOs [85]. Ghana's National Development Planning Commission included NDCs climate change actions in their medium-term plans, requiring ministries and local governments to also do it in their annual plans and budgets [85].

Each of these cases and many others provides a fertile landscape for comparative analysis that can continue to test what institutional and other factors impact cross-scale knowledge production and use.

*6.2. Equity*

Three of the drivers of inequity described above seem particularly well-suited to examine through lenses of salience and legitimacy: disadvantaged communities are often more vulnerable to negative impacts of human and natural disasters; recovery efforts often exacerbate pre-existing inequities; and benefits of sustainable transitions are not always equitably distributed. In each of these conditions, the boundary between vulnerable populations and decision-makers or knowledge-producers is difficult to bridge because of traditional and legacy barriers to access, resources, expertise and political power. In addition, there is often distrust between these populations and decision-makers/knowledge-producers. These challenges are no different *in kind* than those facing general challenges of linking knowledge to action, but given how issues of equity intersect more deeply with issues of race, income inequality, and stark differences of political and economic power, they provide deeper challenges to the interface of science and policy.

Assuring salience and legitimacy can advance equity in several ways. If done correctly, those who are most vulnerable or disadvantaged will have a "seat at the table". This increases the probability of several key drivers of equity. First, being able to help scope and frame the right research and assessment questions will better assure salience—that knowledge produced will best fit the needs of communities that are already vulnerable and answer the questions about which they are most concerned. Second, a seat at the table will increase the chance that stakeholders will see the knowledge-action process as more transparent and inclusive and therefore more legitimate: "without legitimacy, knowledge is unlikely to be trusted. The legitimacy of the knowledge that is produced cannot be assumed by researchers but must be actively sought out and earned" [10] (p. 116).

This suggests two related hypotheses: (1) unless members of vulnerable communities or historically disadvantaged populations are engaged, equity will not be addressed, and (2) special attention to constructing processes that assure salience and legitimacy will increase the chance that equitable outcomes will result. One of these processes could be Joint Fact-Finding (JFF), which explicitly and intentionally allows stakeholders to agree on the question of interest; the way(s) to answer it and the amount of data that is required; how information will be analyzed and interpreted; and how to share what is discovered with those who may be affected by the actions resulting from the answers obtained [86].

### 6.3. Digital Transformation

As radical as the digital revolution has been, there is no *a priori* reason to assume that the framework of salience, credibility and legitimacy is not an effective lens through which to understand how knowledge and action interact in a digital world. In fact, some characteristics of a digital world seem to map well to drivers that enhance salience, credibility and legitimacy: knowledge production and dissemination may be made cheaper, easier, more widespread and more equitably distributed; demand-driven knowledge is more practical; knowledge creation processes can be more transparent; information that informs government decision-making can be more transparent; and digital technologies can enhance co-production.

But, as Lember et al. [67] suggest, the digital revolution is a double-edged sword, in which: data can be increasingly concentrated or centralized either in government or the private sector; expensive technology may be needed to collect and utilize data; new capacity and expertise is needed; data collection and analysis can be more automated and thus less likely to result in co-production; decision-making can be driven through algorithms that remove human interaction; transparency can be decreased and more easily hidden; and inequities are further exacerbated.

These dynamics can diminish salience, credibility and legitimacy for various audiences or stakeholders and thus breed distrust, acting as barriers to knowledge-informed decision-making. These divergent impact paths can both enhance or impede co-production and action driven by knowledge, highlighting the opportunity that strategic and intentional efforts to build salience, credibility and legitimacy may hold in a rapidly changing digital world. Clearly, such efforts would need to be tailored for particular technologies and how they are utilized by knowledge creators and users. This suggests the following hypothesis: like the history of other emerging technologies (e.g., modeling, satellite monitoring, etc.), the deployment of digital technologies does not necessarily secure salience, credibility and/or legitimacy, but acts in ways that can be used to take scientific and technical knowledge either closer or further away from policy/action.

### 6.4. Post-Truth

Of the four stressors described in this paper, this seems to be one in which the applicability of the model of salience, credibility and legitimacy itself is called into question. The criterion of credibility is governed by norms – perhaps norms that might be specific to particular stakeholders or stakeholder groups, but norms nonetheless. From a prescriptive perspective, establishing better outcomes of linking knowledge to action requires understanding those norms and creating institutions

and processes that enhance credibility in a way that decision-makers are more likely to trust the knowledge and use it. To this end, academia created and maintains peer review, the IPCC has developed and evolved mechanisms to review its assessments, government agencies have robust reviews of scientific information, scientific advisory boards have developed clear rules on how and under what conditions data is collected and managed.

In a post-truth world, it is unclear whether scientific credibility has purchase in decision-making and therefore what norms, if any, establish and maintain scientific credibility. In areas where these norms are written into statute or regulations, they are more difficult to erode. In the United States, judiciary system may also protect credibility, as has been demonstrated in a number of lawsuits against executive agencies attempting to diminish the statutory role of science in policy making [87,88]. In more informal and less explicit areas, providing a bulwark against the erosion of credibility has proven more difficult. This is seen, for example, in President Trump's diminishment of the credibility of the public health and epidemiologic response to COVID-19, causing members of the public to take pandemic risks less seriously and act in ways commensurate with that perception of lower risk [89].

In this environment, one hypothesis seems appropriate: the more formally norms to establish credibility are codified (in law, regulations, etc.), the more likely it will be that credibility will be maintained. Despite trustworthiness and community ties degrading in different places over the last decades [10], the COVID-19 crisis may present an unfortunate opportunity to test this hypothesis and ask if there is room for enhanced credibility because of expanded governance of public health issues, like the response to the pandemic. In the long run, we should be able to study the contribution to public trust on scientific knowledge resulting from regulations that explicitly mandate reliance on research centers to track cases and provide real-time information to decide lockdowns and reopening [90], or the impacts on credibility to scientific knowledge from agreements and contracts binding governments and universities to work on treatments and vaccines.

In addition, perhaps the very dynamics that are driving toward a "post-truth" landscape expose important interactions between credibility and legitimacy that have not been adequately explored as of yet. Clearly, concerns about legitimacy driven by the lack of transparency, the use of difficult to track funding for industrial research and direct attacks on the legitimacy of government scientific agencies all can contribute to decreased credibility of science. What are the mechanisms of these interactions? Are there institutional mechanisms that specifically can address such interactions?

## 7. Conclusions

The framework of salience, credibility and legitimacy in co-producing knowledge for action has proven to be a robust model in many contexts, for many different sustainability issues, and in many different places around the world [10]. In the face of new shocks to a rapidly changing world of information, knowledge and technology, might this model remain robust and useful to knowledge producers and knowledge users?

At a time when global governance is increasingly more explicitly linked to local governance, science and action, institutionalizing processes that co-produce knowledge across scales has proven effective. Such processes increase the probability that salience, credibility and legitimacy are established at appropriate levels to create durable action, although there is increasing focus on how the setting of knowledge governance impacts the way in which co-produced knowledge is received and utilized, particularly around challenges appearing at higher scales of decision-making.

With a growing focus on equity and justice as critical parts of sustainable development, a salience, credibility and legitimacy heuristic, if implemented strategically, could result in empowering communities and decision-making that heretofore have been disregarded. The fact that the model explicitly incorporates salience and legitimacy, two criteria without which equity can more easily be ignored, suggests utility in addressing some of the most politically and morally challenging and contentious issues of resource allocation.

The two stressors of rapidly transforming digital knowledge and a "post-truth" world create novel challenges to a co-production model that relies on salience, credibility and legitimacy. In a knowledge landscape driven by AI and algorithms, how do we conceptualize legitimacy and how do we identify knowledge producers and users? In a "post-truth" world in which science itself is questioned and norms of creating legitimate and credible science are disregarded, ideology and politics, which have always been drivers of decision-making [91], become now of overwhelming importance to it.

To address all of these stressors, perhaps widening the scope of analysis may be fruitful. Further research is needed to identify how stressors relate to each other and the consequences of these interactions. For example, the integration of local and global knowledge can potentially transform formerly invisible vulnerable populations into beneficiaries of regional and national policies and plans. On the other hand, we see how the widespread use of digital technologies in the form of social media is amplifying misinformation and creating less trust in science, which may further undermine the ability to link knowledge and action. The interaction of these stressors can cause roadblocks, but also open up latent pathways to more effective knowledge-driven policy action.

We are in the midst of dramatic transformations in our knowledge-action landscape, ones that could not have been predicted two decades ago, and we cannot predict what new stressors will evolve two decades from now. As the urgency of better linking knowledge and action grows, as all indications of the near- to mid-term suggest, models that can both explain and prescribe improved linkages are ever more valuable. One such model that underscores co-production and attention to the salience, credibility and legitimacy of knowledge has proven useful. There are indications that this model may be useful in situations defined by dynamic shifts in the knowledge-action landscape itself.

**Author Contributions:** D.W.C. contributed conceptualization of the work and composition of the article. P.G.B. contributed to the literature review and composition of the article. All authors have read and agreed to the published version of the manuscript.

**Funding:** This research received no external funding.

**Conflicts of Interest:** The authors declare no conflict of interest.

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
