# Peer review of "Salience, Credibility and Legitimacy in a Rapidly Shifting World of Knowledge and Action"

_sustainability, doi:10.3390/su12187376_

Round 1
Reviewer 1 Report
It was a pleasure to read this paper, as I think it is very timely and thought-provoking. As someone that uses the 'salience, credibility, and legitimacy' framing in my own work, I think it does an excellent job of outlining four challenges that are at least somewhat emergent in nature.
While recommending acceptance 'in present form', a few comments/ suggestions that you may want to consider:
1) I was certainly left wanting more. While the hallmark of a good framing paper intended to stake out some important new hypotheses, I wonder if you could go a step further in outlining (in very general terms) a research agenda(s) to test these hypotheses? You explicitly claim, for example, that this hypothesis is testable: "Greater explicit calls to institutionalize linking local to global scales will result in increased salience, credibility and legitimacy of knowledge co-produced across scales and therefore greater connection between knowledge and action from the global to the local level." How might we go about that?
2) On a similar note, aside from research for the sake of hypothesis testing, I wonder if you might point towards approaches that could help to mitigate these tensions. For example, on the equity front you posit that "special attention to constructing processes that assure salience and legitimacy will increase the chance that equitable outcomes will result." What might these look like? Presumably, best-practices in collaborative governance, including potentially the use of techniques like joint fact-finding, could be pointed towards (?).
3) With the 'digital transformation', I think it is implicit but you might place more emphasis on the tradeoffs associated with 'big data' itself. I would suggest that data analytics can provide invaluable information for decision-making and, as you suggest, credibility and legitimacy can be increased when, among other things, the data is transparent. However, analytical 'black boxes' can disempower stakeholders by generating 'the answer' rooted in assumptions and lacking in transparency and adequate deliberation around design, data selection, etc.
4) In some ways, the 'post-truth' paradigm we may increasingly find ourselves in seems the most paralyzing, at least to me. You point towards ways in which we might overcome the other three, but this one seems the most vexing when communities of interest can simply disregard or hostilely view what is viewed as critical data by others. Similar to above, I wonder what "more formal[al..] norms to establish credibility" might be. I'm honestly most skeptical of this hypothesis but seems very worthy of testing.
5) Line 307: assume you mean 'revolution' not 'resolution'
Author Response
RESPONSES ARE IN ITALICS
It was a pleasure to read this paper, as I think it is very timely and thought-provoking. As someone that uses the 'salience, credibility, and legitimacy' framing in my own work, I think it does an excellent job of outlining four challenges that are at least somewhat emergent in nature.
While recommending acceptance 'in present form', a few comments/ suggestions that you may want to consider:
1) I was certainly left wanting more. While the hallmark of a good framing paper intended to stake out some important new hypotheses, I wonder if you could go a step further in outlining (in very general terms) a research agenda(s) to test these hypotheses? You explicitly claim, for example, that this hypothesis is testable: "Greater explicit calls to institutionalize linking local to global scales will result in increased salience, credibility and legitimacy of knowledge co-produced across scales and therefore greater connection between knowledge and action from the global to the local level." How might we go about that?
In some sections we added more specific outlines for a research agenda, including comparative case studies in the section on scales.
2) On a similar note, aside from research for the sake of hypothesis testing, I wonder if you might point towards approaches that could help to mitigate these tensions. For example, on the equity front you posit that "special attention to constructing processes that assure salience and legitimacy will increase the chance that equitable outcomes will result." What might these look like? Presumably, best-practices in collaborative governance, including potentially the use of techniques like joint fact-finding, could be pointed towards (?).
Thank you for the reminder of the joint fact-finding literature. We neglected this in the original and added something in the revised version.
3) With the 'digital transformation', I think it is implicit but you might place more emphasis on the tradeoffs associated with 'big data' itself. I would suggest that data analytics can provide invaluable information for decision-making and, as you suggest, credibility and legitimacy can be increased when, among other things, the data is transparent. However, analytical 'black boxes' can disempower stakeholders by generating 'the answer' rooted in assumptions and lacking in transparency and adequate deliberation around design, data selection, etc.
We added a portion to this section addressing this concern.
4) In some ways, the 'post-truth' paradigm we may increasingly find ourselves in seems the most paralyzing, at least to me. You point towards ways in which we might overcome the other three, but this one seems the most vexing when communities of interest can simply disregard or hostilely view what is viewed as critical data by others. Similar to above, I wonder what "more formal[al..] norms to establish credibility" might be. I'm honestly most skeptical of this hypothesis but seems very worthy of testing.
Another reviewer pushed us to look at the interaction of legitimacy and credibility that we believe responds to this comment. We added text to address this.
5) Line 307: assume you mean 'revolution' not 'resolution'
Changed
Reviewer 2 Report
I want to congratulate the authors. They produced an interesting, well-written and well documented manuscript. One of the key missing features of co-constructing is how to do it effectively and in the long-run, that is, we have had for a long time a perennial deficit of substantive democracy. Although the authors refer sources of information, the reader may wonder about the how to, that is, at least some details, about the methodologies for operationalizing the collaborative processes proposed. However, such component may require another paper.
Author Response
RESPONSES ARE IN ITALICS
I want to congratulate the authors. They produced an interesting, well-written and well documented manuscript. One of the key missing features of co-constructing is how to do it effectively and in the long-run, that is, we have had for a long time a perennial deficit of substantive democracy. Although the authors refer sources of information, the reader may wonder about the how to, that is, at least some details, about the methodologies for operationalizing the collaborative processes proposed. However, such component may require another paper.
We agree that a detailed exposition of specific methodologies would require a separate paper, the other reviewers had similar responses and we added several specific areas of exploration of methodologies.
Reviewer 3 Report
This paper presents a review of recent trends that shape science policy dynamics and in particular the relevance of the well established credibility, salience, legitimacy criteria. It is well written, timely, and interesting to read, so I would suggest that with some revision it is suitable for publication in this journal and would be of interest to the readership. I detail some major and minor comments below.
Major comments
The paper could be read as an uncritical reconfirmation of an existing framework - it went to assess the relevance of the CSL criteria and low and behold found they are relevant again. I would encourage the authors to be a little more critical than they have been and also to consider in their reflections whether there are other criteria or aspects of knowledge co-production that may need to be situated alongside the criteria. I don’t have suggestions as to what they may be, but it seems like a useful exercise to contemplate. Further to this point, the authors have suggested some ‘testable hypotheses’ but have not really included a discussion on what it might look like to test them and therefore what the future research and practical next steps and outcomes of this paper might be. The CSL criteria have had immense practical utility for people working at the interface of science policy and practice, and I would encourage the authors to reflect on what this thought piece means for how they go about their work in a world shaped by these forces. This could take the form of reflection in the conclusion, or along side the testable hypothesis, some speculation guidance for practitioners.
The discussion is largely silent on the institutional barriers to knowledge use that are not addressed by the CSL criteria, I would assume that the cross-scale section is the most relevant here, but there is increasing focus on ways in which the governance setting of knowledge co-production/knowledge use shapes the way in which knowledge is received and utilized. We know a lot about what makes for good co-production, in part inspired by the CSL criteria, but these processes continue to face challenges that stem from higher scales of decision-making/power than the site of engagement, there is insufficient scholarship and reflection on how those challenges can be addressed (see work on knowledge governance, van Kerkhoff 2013, also discussed in Wyborn et al 2019). A focus on cross scale context seems to be an opportunity here to point to a need to look at the ways in which these dynamics enhance or undermine the use of knowledge whether it is co-produced or not.
Suggest that in the conclusion it would be worth reflecting on the intersections between forces and their implications for the CSL criteria?
6.4 - post truth - is there something about legitimacy to be explored here? I’m not quite sure what the hypothesis is but seems to me that part of the issue here is the lack of transparency around the ‘dark money’ that sponsors many of the attacks on science that have been so profound. Perhaps to ‘happy clappy’ a recommendation here, but is there something to be considered around the role of transparency/legitimacy in generating credibility. I don’t know. Perhaps this assumption falls into the trap of suggesting that if a process is open and the science is well communicated then people will ‘buy into it’ which clearly ignores what has been shown about the role of values in the way in which people engage with scientific information (which, as I write, I think is probably something else that could be included in the post-truth section). I don’t know what the appropriate response is here, but I just feel that singularly focusing this section on credibility misses some of the broader context of what is driving some of the post-truth politics.
Line 367/368: I would suggest that politics has always been a driver of decision making, and that is actually quite appropriate in a democracy, what has changed now is that the nature of that politics has made it even more difficult for evidence to be taken as a viable input into decision-making. But even before the post-truth era the idea that evidence rather than values should drive decision making was problematic. Sarewitz ‘why science makes environmental controversies worse’ comes to mind here.
Minor comments
needs to be read for typos related to spacing - some places where spaces are missing and others where double spaces are placed in the middle of sentences
section 3 - page 4 lines 155-157 - Also relevant here is the question of how to processes of knowledge (co)production empower particular voices and knowledges? Also, I would think this is not just about the transparency and legitimacy of how knowledge is created but the politics of knowledge (co)production process and their role in shaping who wins, who loses, and which knowledge is privileged. And typo, the h of how fair should be capitalized
table 1 - resources/processing cell: typo? should it be increase or decrease?
Page 6 line 202. Suggest providing your own concluding sentence here, seems sloppy to end with direct quote of another paper.
Author Response
Thank you! We found these suggestions very helpful!
RESPONSES ARE IN ITALICS
Major comments
The paper could be read as an uncritical reconfirmation of an existing framework - it went to assess the relevance of the CSL criteria and low and behold found they are relevant again. I would encourage the authors to be a little more critical than they have been and also to consider in their reflections whether there are other criteria or aspects of knowledge co-production that may need to be situated alongside the criteria. I don’t have suggestions as to what they may be, but it seems like a useful exercise to contemplate. Further to this point, the authors have suggested some ‘testable hypotheses’ but have not really included a discussion on what it might look like to test them and therefore what the future research and practical next steps and outcomes of this paper might be. The CSL criteria have had immense practical utility for people working at the interface of science policy and practice, and I would encourage the authors to reflect on what this thought piece means for how they go about their work in a world shaped by these forces. This could take the form of reflection in the conclusion, or along side the testable hypothesis, some speculation guidance for practitioners.
Good point. We added some acknowledgement of critiques.
The discussion is largely silent on the institutional barriers to knowledge use that are not addressed by the CSL criteria, I would assume that the cross-scale section is the most relevant here, but there is increasing focus on ways in which the governance setting of knowledge co-production/knowledge use shapes the way in which knowledge is received and utilized. We know a lot about what makes for good co-production, in part inspired by the CSL criteria, but these processes continue to face challenges that stem from higher scales of decision-making/power than the site of engagement, there is insufficient scholarship and reflection on how those challenges can be addressed (see work on knowledge governance, van Kerkhoff 2013, also discussed in Wyborn et al 2019). A focus on cross scale context seems to be an opportunity here to point to a need to look at the ways in which these dynamics enhance or undermine the use of knowledge whether it is co-produced or not.
We added points on this and cited van Kerhoff’s work.
Suggest that in the conclusion it would be worth reflecting on the intersections between forces and their implications for the CSL criteria?
6.4 - post truth - is there something about legitimacy to be explored here? I’m not quite sure what the hypothesis is but seems to me that part of the issue here is the lack of transparency around the ‘dark money’ that sponsors many of the attacks on science that have been so profound. Perhaps to ‘happy clappy’ a recommendation here, but is there something to be considered around the role of transparency/legitimacy in generating credibility. I don’t know. Perhaps this assumption falls into the trap of suggesting that if a process is open and the science is well communicated then people will ‘buy into it’ which clearly ignores what has been shown about the role of values in the way in which people engage with scientific information (which, as I write, I think is probably something else that could be included in the post-truth section). I don’t know what the appropriate response is here, but I just feel that singularly focusing this section on credibility misses some of the broader context of what is driving some of the post-truth politics.
Excellent point here and we made significant additions to address this.
Line 367/368: I would suggest that politics has always been a driver of decision making, and that is actually quite appropriate in a democracy, what has changed now is that the nature of that politics has made it even more difficult for evidence to be taken as a viable input into decision-making. But even before the post-truth era the idea that evidence rather than values should drive decision making was problematic. Sarewitz ‘why science makes environmental controversies worse’ comes to mind here.
Good suggestion. We noted this dynamic.
Minor comments
needs to be read for typos related to spacing - some places where spaces are missing and others where double spaces are placed in the middle of sentences
section 3 - page 4 lines 155-157 - Also relevant here is the question of how to processes of knowledge (co)production empower particular voices and knowledges? Also, I would think this is not just about the transparency and legitimacy of how knowledge is created but the politics of knowledge (co)production process and their role in shaping who wins, who loses, and which knowledge is privileged. And typo, the h of how fair should be capitalized
We added a point on this.
table 1 - resources/processing cell: typo? should it be increase or decrease? fixed
Page 6 line 202. Suggest providing your own concluding sentence here, seems sloppy to end with direct quote of another paper. Fixed
Round 2
Reviewer 3 Report
The authors have adequately addressed my comments and suggestions. They are to be commended for a very interesting paper that I'm sure will stimulate more discussion and debate.
Thank you for the opportunity to review this paper.